# A Synthetic Cell-Penetrating Heparin-Binding Peptide Derived from BMP4 with Anti-Inflammatory and Chondrogenic Functions for the Treatment of Arthritis

**DOI:** 10.3390/ijms21124251

**Published:** 2020-06-15

**Authors:** Da Hyeon Choi, Dongwoo Lee, Beom Soo Jo, Kwang-Sook Park, Kyeong Eun Lee, Ju Kwang Choi, Yoon Jeong Park, Jue-Yeon Lee, Yoon Shin Park

**Affiliations:** 1School of Biological Sciences, College of Natural Sciences, Chungbuk National University, Cheongju 28644, Korea; dahyun6682@hanmail.net (D.H.C.); dlruddms1223@naver.com (K.E.L.); jukwang@chungbuk.ac.kr (J.K.C.); 2Central Research Institute, Nano Intelligent Biomedical Engineering Corporation (NIBEC), School of Dentistry, Seoul National University, Seoul 03080, Korea; ldw410@nibec.co.kr (D.L.); beomsoojo@nibec.co.kr (B.S.J.); parkyj@snu.ac.kr (Y.J.P.); 3Department of Dental Regenerative Bioengineering and Dental Research Institute, School of Dentistry, Seoul National University, Seoul 03080, Korea; soooook7424@snu.ac.kr

**Keywords:** anti-inflammatory peptide, cell permeable peptide, heparin-binding peptide, collagen-induced arthritis, inducible nitric oxide, interferon gamma, interleukin-6, Enbrel

## Abstract

We report dual therapeutic effects of a synthetic heparin-binding peptide (HBP) corresponding to residues 15–24 of the heparin binding site in BMP4 in a collagen-induced rheumatic arthritis model (CIA) for the first time. The cell penetrating capacity of HBP led to improved cartilage recovery and anti-inflammatory effects via down-regulation of the iNOS-IFN*γ*-IL6 signaling pathway in inflamed RAW264.7 cells. Both arthritis and paw swelling scores were significantly improved following HBP injection into CIA model mice. Anti-rheumatic effects were accelerated upon combined treatment with Enbrel^®^ and HBP. Serum IFN*γ* and IL6 concentrations were markedly reduced following intraperitoneal HBP injection in CIA mice. The anti-rheumatic effects of HBP in mice were similar to those of Enbrel^®^. Furthermore, the combination of Enbrel^®^ and HBP induced similar anti-rheumatic and anti-inflammatory effects as Enbrel^®^. We further investigated the effect of HBP on damaged chondrocytes in CIA mice. Regenerative capacity of HBP was confirmed based on increased expression of chondrocyte biomarker genes, including aggrecan, collagen type II and TNFα, in adult human knee chondrocytes. These findings collectively support the utility of our cell-permeable bifunctional HBP with anti-inflammatory and chondrogenic properties as a potential source of therapeutic agents for degenerative inflammatory diseases.

## 1. Introduction

Rheumatic arthritis (RA) is one of the most common autoimmune rheumatic diseases (AIRD) caused by dysregulation of tolerance to self-antigens, leading to chronic systemic inflammatory disorders involving the musculoskeletal system [1]. According to Korean National Health Insurance (NHI) Claims Database reports, the steadily increasing prevalence of RA poses a significant economic burden [2]. 

The causes and mechanisms of action of RA have been relatively unexplored to date. The primary treatment goal is control of joint inflammation and reduction of joint damage, the major determinant of functional prognosis in RA [3]. The most common therapeutic agents for RA are disease-modifying anti-rheumatic drugs (DMARD), traditionally comprising synthetic chemicals and small molecules that act as anti-inflammatory agents [4,5]. 

TNFα is a key cytokine in the inflammatory process in RA [6]. Five TNFα blockers have been approved by US-FDA for RA therapy, specifically, adalimumab, certolizumab pegol, etanercept, infliximab, and golimumab. In randomized controlled clinical trials, these drugs are reported to be effective in recovery of inflammation-related clinical signs in RA patients with response failure to synthetic DMARDs [7]. Multiple studies have demonstrated remarkable therapeutic benefits of early co-treatment with TNFα blockers and methotrexate [7]. Other FDA-approved drugs for treatment of moderate to severe RA include abatacept, rituximab, and tocilizumab. However, these drugs commonly induce severe side-effects, such as osteoporosis, liver function failure, and even lymphoma, due to the complexity of the inflammatory network [8,9]. Various attempts have been made to counteract the side-effects of these therapeutic agents [10]. Treatment with synthetic disease-modifying anti-rheumatic drugs (DMARD), including methotrexate, sulfasalazine, and leflunomide, represents an important paradigm shift that can lead to remarkable improvement of clinical symptoms and delayed joint damage. Despite the effectiveness of these medications, a significant number of RA patients continue to experience clinical symptoms of inflammation and progressive joint destruction, clearly suggesting that therapeutic blockade with any one cytokine is not necessarily sufficient for relief of RA [11,12]. 

To improve therapeutic efficacy and/or prolong drug persistence, various tissue-engineered materials have been developed. For instance, an injectable electrostatically interacting drug depot [13] and click-crosslinked hyaluronic acid depot [14] have been developed as a supplementary or combination therapeutic agent for RA with minocycline and methotrexate, respectively. Direct intra-articular injection resulted in increased duration of therapy and enhanced relief of symptoms by inducing sustained release of the ionic drug. Additionally, novel biomimetic scaffolds or drug carriers have been suggested as modes for delivering nanosized hydroxyapatite (HA) to promote damaged tissue recovery [15] through osteoblast adhesion [16] and bone adjuvant activities [17]. Limitations of tissue-engineered agents or scaffolds are mostly attributed to uncontrolled proinflammatory cytokines and inflammation-related signaling during RA pathogenesis. 

Biological compounds, such as neutralizing antibodies against interleukin-17A (IL17A; secukinumab and ixekizumab) and janus kinase (JAK) inhibitors (tofacitinib, baricitinib, GLPG0634, and VX-509) [18,19], have been characterized as beneficial agents with significant therapeutic effects. In particular, IL6 neutralization is a recently developed therapeutic strategy for controlling RA. Neutralization of IL6 plays a critical role in both the initiation and perpetuation of immunologic dysfunction and inflammatory responses in RA via modulation of the IL6/IL6 receptor/gp130 pathway [20]. A novel role of IL6/IL6R/gp130 signaling in autoimmune arthritis has been reported. An apparent strong link exists between enhanced Toll-Like Receptor (TLR) expression as a consequence of RA development and production of IL6 accompanied by upregulation of MMP genes [21,22,23]. However, protein-based biological agents are expensive and associated with safety concerns, such as development of opportunistic and viral infections [24,25,26]. Biological agents, such as neutralizing antibodies that are mainly composed of protein polymers, can induce severe infections, drug resistance, and/or autoimmune reactions unless completely degraded in the body [27,28]. 

To overcome these limitations, the use of peptide drugs as substitutes for common RA therapeutics has been widely investigated in recent years [18,19,29]. The advantages of peptide agents include high selectivity, high potency, low immunogenicity, low toxicity, improved efficiency and safety, low tissue accumulation, easy combination with biomaterial surfaces via chemical modifications, and low cost [19,30]. Safe and effective novel peptide therapeutic agents for RA treatment are urgently required.

Here, we focused on the anti-inflammatory effects of a synthetic heparin-binding peptide (HBP) composed of 10 amino acids derived from the heparin binding site of bone morphogenetic protein 4 (BMP4) [31]. The HBP sequence examined in this study, RKKNPNCRRH, corresponds to residues 15–24 of the heparin binding site in BMP4, and is similar to the HBP “consensus sequence” BBBXTXXBBB (whereby X, B, and T represent hydropathic residue, basic residue, and turn, respectively) [31]. We previously identified an anti-angiogenic role of HBP in endothelial cells (EC) and a breast cancer xenograft model [32]. The collective results suggest that HBP may serve as potential antitumor agents by regulating the tumor microenvironment through interactions with several cytokines [32]. Human heparin binding domain is the most recently discovered member of the host defense peptide family involved in adaptive immunity and anti-inflammatory activity [33,34]. The domain itself is unable to penetrate immune cells for chemical production due to its high amino acid content. However, selected key peptide sequences of this domain with cell penetrating activity have successfully achieved inhibition of intracellular inflammatory signaling in cells. 

The main objective of this study was to investigate the potential anti-inflammatory and chondrocyte regenerative effects of a cell permeable HBP, RKKNPNCRRH, derived from the heparin binding site of BMP4 in lipopolysaccharide (LPS)-treated macrophages and human articular chondrocytes in vitro and a CIA-induced RA mouse model in vivo. 

## 2. Results

### 2.1. Characterization of HBP

The primary and secondary structures of the synthetic HBP derived from the human heparin binding domain are presented in Figure 1A,B. The HBP sequence was composed of 10 amino acids (RKKNPNCRRH, Figure 1A). The proline acts as a turning site and induces flexibility to the structure, which facilitates penetration of cells or tissues. The molecular mass of HBP was determined as 1307.7087, which is the optimal size for ease of cell penetration (Appendix A). Additionally, the positive surface charge of HBP (net charge, +5) enhanced cell penetration capacity. Secondary structures are presented in ball-and-stick and cartoon styles (Figure 1B).

### 2.2. Cellular Uptake Activity of HBP

The cell penetration capacity of HBPs was examined by treatment of RAW264.7 cells with a range of HBP concentrations for 10 min, followed by confocal microscopy (Figure 2A). As shown in Figure 2A, rhodamine-labeled HBP penetrated the cytoplasm and even nucleus of RAW264.7 cells. The penetration capacity was increased in a concentration-dependent manner, with the highest infiltration observed at 100 µg/mL HBP. Our results support the utility of the newly synthesized HBP from BMP4 as a delivery agent for large therapeutic molecules, such as DNA, RNA, antibodies, and proteins, with minor safety concerns owing to its derivation from human protein. We further examined cell penetration capacity under different temperatures (4 and 37 °C) and incubation times (10 min, 1 h, and 4 h) (Figure 2B). HBP penetration of RAW264.7 cells at 4 °C was observed at time-points of 10 min (pink line) and 1 h (orange line), but not 4 h (yellow line), which may be attributed to decreased cell viability due to long-term exposure to low temperature. Penetration was improved in cells incubated at high temperature for a longer period (37 °C for 4 h) (Figure 2B) and evident after only 10 min of incubation. Penetration capacity was therefore slightly increased with temperature in a time-dependent manner (Figure 2B). Cell uptake of HBP occurs via both direct penetration and endocytosis.

### 2.3. Anti-Inflammatory Effects of HBP on LPS-Treated RAW264.7 Cells

Upon treatment of RAW264.7 cells with LPS (1 μg/mL) for 24 h, their unique bubble-like shape altered to a fibroblast-like morphology, indicative of stimulation of the inflammatory response (Figure 3A, pre-HBP treatment). Treatment of LPS-stimulated cells with HBP (100 μg/mL) for 1 h led to recovery of the unique morphology of RAW264.7 cells (Figure 3A, post-HBP treatment) (Figure 3B,C).

### 2.4. Effects of HBP on Proteins Related to the Inflammation Pathway

To further confirm the anti-inflammatory activity of HBP, LPS-stimulated RAW264.7 cells were treated with varying concentrations of peptide (0, 10, 50, and 100 μg/mL) for 24 h, and changes in levels of inflammation-related proteins, including iNOS (Figure 4A,B), COX2 (Figure 4A,C), IFN*γ* (Figure 4A,D), and IL6 (Figure 4A,E), examined in cell lysates. Compared to the non-treated group (NT), iNOS, COX2, IFN*γ*, and IL6 protein levels presented as image densities were significantly increased following LPS treatment (62.4-, 28.6-, 1.2-, and 4.6-fold increase compared to the NT group, respectively). Inflammatory protein expression was significantly suppressed by HBP in a dose-dependent manner. At the highest HBP concentration (100 μg/mL), iNOS, COX2, IFN*γ*, and IL6 protein levels were significantly decreased to 0.41-, 0.74-, 0.30-, and 0.13-fold, respectively, relative the value of 1.0 at 0 μg/mL HBP. IL6 expression was the most significantly suppressed by HBP.

### 2.5. Chondrocyte Recovery Effect of HBP in Human Articular Chondrocytes

We first evaluated the effect of HBP on NHAC cells without LPS stimulation to clarify the chondrogenic potential of the peptide alone (Figure 5A–C). To determine whether our newly synthesized HBP could affect recovery of chondrocytes, aggrecan (AGG; Figure 5D), collagen type II (COLII; Figure 5E), and TNFα (Figure 5F) gene expression changes were evaluated in LPS-stimulated chondrocytes after 5 days of HBP treatment (Figure 5). Quantitative RT-PCR analyses revealed that LPS stimulation suppressed AGG and COLII and enhanced TNFα expression. HBP treatment induced significant recovery of AGG and COLII expression (0, 10, 50 and 100 μg/mL), but had a slight and not significant effect on TNFα expression. In view of the HBP-mediated recovery of damaged chondrocytes, we suggest that the peptide improves chondrocyte-specific characteristics through effects on AGG, COLII, and TNFα, even under inflammatory conditions.

### 2.6. Antiarthritic Effects of HBP on CIA Mice

#### 2.6.1. Hind Paw Swelling, Arthritis Score, and Histological Recovery of CIA Mice Injected with HBP

Compared with the normal control group (NT), the CIA control group (PBS) showed significant hind paw swelling. An experimental scheme depicting HBP activity in the CIA mouse model is presented in Figure 6A. Enbrel^®^, an agent widely used to treat rheumatoid arthritis in the clinic, was selected as a positive control [35]. Intraperitoneal injection of HBP and Enbrel^®^ was performed into the lower right quadrant of the abdomen of mice for comparison of therapeutic efficacy. 

CIA mice displayed significantly swollen hind paws, which was confirmed with visual evaluation (Figure 6B, #1 and #2 panel). The anti-rheumatic effect of HBP in CIA mice was histologically evaluated (Figure 6C). The CIA-PBS group developed chronic inflammation in the synovial tissue, with synovial proliferation (sp) and pannus formation (yellow arrow, cartilage and bone destruction, and bone erosion) (Figure 6C-b), compared to the NT group (Figure 6C-a). The Enbrel^®^-treated CIA group displayed decreased inflammation to a lower extent than the HBP-treated CIA group (Figure 6C-c). Inflammation was markedly reduced in the HBP-treated CIA group (Figure 6C-d). Synovial cell hyperplasia, bone erosion, and pannus formation were additionally attenuated in groups treated with HBP. Recovery of cartilage morphology in the HBP-treated group was similar to that in the NT and Enbrel^®^ treatment groups (Figure 6C-a,e).

Hind paw swelling score was notably increased in CIA mice (8.99 ± 0.99, *p* < 0.05), compared to the NT group (6.54 ± 0.09), at the end of the experimental period (day 28) (Figure 6D). Relative to the CIA-PBS group, the Enbrel^®^-treated CIA group showed significantly reduced hind paw swelling from days 11 to 28 (*p* < 0.05). The HBP-treated CIA group also presented markedly decreased hind paw swelling on day 11, which was maintained until day 28 (*p* < 0.05). No significant differences between the Enbrel^®^ and HBP treatment groups were evident throughout the experimental period (Figure 6E), indicative of similar anti-rheumatic effects. 

Consistent with data on hind paw swelling, the histological arthritis score in CIA mice was markedly lower following HBP treatment, compared to that in the CIA-PBS group (3.3 ± 0.67, *p* < 0.05; Figure 6F, Enbrel^®^ group; 1.6 ± 0.2). In the HBP and Enbrel^®^ co-injected CIA group, histological arthritis score (2.3 ± 0.3) was significantly decreased relative to that of the CIA-PBS group (3.3 ± 0.67). HBP treatment clearly suppressed inflammatory cell infiltration and reduced the arthritis score in CIA mice (Figure 6G). 

The combined effects of HBP and Enbrel^®^ were examined in CIA mice. The combination of peptide and Enbrel^®^ exerted a marginally greater therapeutic effect on paw swelling than each single agent alone (Figure 6D,E). Arthritis score was improved with HBP as well as Enbrel^®^ injection for 28 days (Figure 6F,G). In general, the degree of recovery of arthritis was further improved upon co-injection with the peptide and Enbrel^®^ both agents.

#### 2.6.2. HBP Induces Suppression of Serum Inflammatory Cytokine Levels

Serum concentrations of IFN*γ* and IL6 in CIA mice were examined to ascertain the therapeutic effect of HBP (Figure 7). IFN*γ* (37.28 ± 11.86 pg/mL) and IL6 (242.77 ± 77.48 pg/mL) levels were significantly increased in CIA mice (RA-PBS group) on day 28, compared to the NT group (8.86 ± 2.29 pg/mL and 98.09 ± 4.32 of IFN*γ* and IL6 concentrations, respectively; *p* < 0.001).

Serum concentrations of IFN*γ* (24.88 ±11.82 pg/mL) and IL6 (158.02 ± 55.09 pg/mL) in Enbrel^®^-treated CIA mice were significantly decreased, compared to CIA-PBS mice. The values obtained from HBP-treated CIA mice were similar to those of the Enbrel^®^ group. Serum levels of IFN*γ* and IL6 were reduced to 27.79 ± 15.63 pg/mL and 198.18 ± 51.63 pg/mL, respectively, following HBP injection (Figure 7A,B). Co-treatment with HBP and Enbrel^®^ additionally induced a marked reduction in serum IFN*γ* and IL6 levels to 23.61 ± 8.98 pg/mL and 161.27 ± 47.96 pg/mL, respectively.

#### 2.6.3. Safranin O-Fast Green Staining and Immunohistochemical Analysis of IL6 in Cartilage of CIA Mice

To further validate the chondrogenic potential of HBP, we performed Safranin O-fast green staining and analyzed chondrocyte recovery using the Image J program (Figure 8A,C). The red-stained glycosaminoglycan (GAG) area was significantly damaged in CIA-induced RA mice, compared with the RA-PBS group (1.0 ± 0.01 vs. 0.21 ± 0.01, NT vs. RA-PBS; *p* < 0.05). Treatment with Enbrel^®^ promoted recovery of damaged chondrocytes (0.79 ± 0.02 vs. 0.21 ± 0.01, Enbrel^®^ vs. RA-PBS; *p* < 0.05). Notably, while the HBP-only treatment group showed significant recovery of chondrocyte regeneration potential relative to the PBS group, co-treatment with Enbrel^®^ was required for optimal results. Our findings imply that HBP is more effective as a co-therapeutic complementing the activity of Enbrel^®^ than as a single treatment agent. 

Immunohistochemical analysis was performed to evaluate HBP-induced changes in IL6 protein expression in CIA mice. Limb joints of male CIA mice were obtained at the end of the experimental period. IL6 expression was significantly higher in CIA mice (RA-PBS, 1.88-fold increase), compared to the NT group (Figure 8B,D). 

Enbrel^®^-treated CIA mice showed a slight reduction in IL6 expression, compared with the RA-PBS group (1.88 ± 0.12 vs. 1.80 ± 0.03, Enbrel^®^ vs. RA-PBS; *p* > 0.05). However, HBP treatment induced significant IL6 reduction relative to the RA-PBS treatment group (1.39 ± 0.03 vs. 1.80 ± 0.03, Enbrel^®^ vs. RA-PBS; *p* < 0.05). Importantly, the most significant decrease in IL6 expression was observed in the co-injection group (E+HBP) (1.12 ± 0.02 vs. 1.80 ± 0.03, E+HBP vs. RA-PBS; *p* < 0.05).

## 3. Discussion

Previously, we synthesized a heparin binding domain peptide (HBP) derived from human heparin binding domain of BMP4 and evaluated its biological significance in tumor targeting and anti-angiogenesis [32]. BMP4 is a multifunctional growth factor that mainly promotes bone formation [36]. This growth factor is expected to be highly efficacious in clinical treatment of various diseases owing to involvement in multiple physiological processes, such as regulation of angiogenesis [32], muscle development, human embryo development, and mineralization of bone [31]. In lieu of BMP4 protein, a peptide composed of 10 amino acids, RKKNPNCRRH, corresponding to residues 15–24 of the heparin binding site of BMP-4 was generated and evaluated for therapeutic application in various diseases. 

We initially examined the efficacy of the BMP-4-derived HBP against RA and whether the peptide could reduce the side-effects of Enbrel^®^, with a view to assessing its utility as a candidate or alternative agent for RA therapy. Additionally, the effects of HBP on chondrocyte recovery were investigated in an RA animal model. 

To trigger an anti-inflammatory effect, peptides need to be efficiently delivered into cells. The amino acid sequence of HBP exhibits similarities with cell-penetrating peptides containing poly-arginine [37]. Cell penetration properties are attributed to eight positive amino acids [38]. The cell penetration activity of the newly developed HBP was evaluated by our group using human breast cancer cells in a preliminary study [32] and RAW264.7 cells and artificial skin tissue in the current study (Figure 2A). The infiltration capacity of HBP into inflamed cells was demonstrated for the first time using an LPS-stimulated in vitro model (Figure 5A). As shown in Figure 2A, rhodamine-tagged HBP was successfully delivered into the cytoplasm and even nucleus of inflammatory RAW264.7 cells. HBP penetration into cells may not be required owing to its heparan sulfate proteoglycan (HSPG) binding affinity. Interactions of inflammatory mediators, such as chemokines, with HSPG are known to modulate the inflammation process. HSPGs play a key role in inflammation and their mimetics effectively act as anti-inflammatory agents through modulating the interactions between mediators and HSPGs [39]. Therefore, the anti-inflammatory activity of HBP may be mediated through binding to HSPGs. In addition to HSPG-binding affinity, HBP displayed cell uptake ability, even at 4 °C, indicative of direct cell penetration (Figure 2B). Intracellular distribution patterns of rhodamine-labeled HBP, observed via confocal microscopy, supported its successful delivery into cells (Figure 2A). Besides bound HSPGs, HBP delivered into cells may affect intracellular signaling pathways, such as NF-kB, involving production of proinflammatory cytokines, including TNF-α and IL6. Further studies are required to establish whether HBP exerts anti-inflammatory effects through binding to HSPGs or inhibition of specific intracellular targets.

Following delivery into inflammatory cells, HBP induced a decrease in expression of inducible nitric oxide synthase (iNOS), cyclooxygenase 2 (COX2), and interferon gamma (IFN*γ*) proteins triggered by the intracellular inflammatory response (Figure 4). Nitric oxide (NO) is closely associated with inflammation, angiogenesis, and tissue destruction in the RA model. Interplay of pro- and anti-inflammatory cytokines promotes iNOS production in the affected tissues of RA patients [40]. The iNOS enzyme is responsible for localized overproduction of NO in synovial joints affected by RA. Poulami et al. [41] suggested that the determination of the cytokine signaling network underlying regulation of iNOS is essential to understand the pathophysiology of RA progression. Microarray data analysis by the group revealed upregulation of the gene network belonging to interferon gamma (IFN*γ*) and interleukin 6 (IL6) pathways in the RA synovium. Conversely, genes contributing to the anti-inflammatory transforming growth factor-beta (TGFβ) signaling pathway were downregulated [42]. In the current study, iNOS and COX2 protein levels were significantly decreased at HBP concentrations of 50 µg/mL to 100 µg/mL (about two-fold), compared to control cells, with limited differences between the 50 and 100 µg/mL treatment groups. Complete recovery from acute inflammation induced by LPS to basal levels within a short time-frame (24 h in the current experiment) may be difficult to achieve. Although HBP successfully exerted an anti-inflammatory effect on LPS-treated RAW264.7 cells, as evident from decreased iNOS and COX2 protein expression, the initial significant increases in iNOS and COX2 induced by LPS were extremely high and could not be reduced to basal levels by HBP. Expression of IFN*γ*, a known representative downstream molecule of iNOS signaling in pre- and pro-inflammatory pathways, was markedly decreased upon treatment with 100 µg/mL HBP. Accordingly, the optimum concentration of HBP for controlling inflammatory reaction was determined as 100 µg/mL, and this concentration was also confirmed by an MTT assay (Appendix A). In this respect, inhibition of the iNOS-IFN*γ* signaling pathway and IL6 may be contributory factors to the anti-inflammatory mechanism of HBP. 

Next, we examined the therapeutic effect of HBP in vivo via intraperitoneal injection twice per week for 4 weeks into collagen-induced induced arthritic (CIA) DBA/1 mice, a common autoimmune disease model widely employed to study rheumatoid arthritis (RA) [43]. 

Enbrel^®^, a tumor necrosis factor (TNF) blocker, is an FDA-approved RA therapeutic agent indicated for reducing signs and symptoms, inducing major clinical responses, inhibiting progression of structural damage, and improving physical function in patients with RA [44]. Serious side-effects of Enbrel^®^ include susceptibility to new infections, induction of hepatitis B, nervous system problems, such as multiple sclerosis, seizures or inflammation of nerves of the eyes, blood problems, heart failure, and allergic or autoimmune reactions. Long-term use of Enbrel^®^ raises further safety concerns, such as lymphoma and other malignancies. Owing to its wide range of side-effects, limited use over short periods of time is recommended, highlighting the need to develop new, effective therapies to replace or supplement Enbrel^®^ [44]. Here, we compared the therapeutic effects of HBP and Enbrel^®^ in the CIA model. The Enbrel^®^ dose was obtained from a previous report by our group, where CIA mice received 100 µg Enbrel^®^ intraperitoneally (5 mg/kg) twice per week for 4 weeks [45]. 

Severe RA is associated with deformation of joints, along with cartilage and bone destruction (Figure 6), which is the most representative symptom [46,47,48]. In the current study, bone structure and articular chondrocytes at the joint in our mouse model were recovered upon HBP treatment (Figure 6). CIA mice treated with Enbrel^®^ showed significant improvements in arthritis and paw swelling scores (Figure 6D–G). Moreover, damaged chondrocytes in CIA mice detected via H&E staining were significantly reduced following Enbrel^®^ treatment (Figure 6C).

Arthritis and hind paw swelling scores were not significantly different between Enbrel^®^ and HBP-treated groups throughout the experimental period, implying that the HBP exerts similar therapeutic effects as Enbrel^®^ (Figure 6). 

To further elucidate the combined effects of HBP and Enbrel, mice were simultaneously co-injected with both agents. Co-injection induced a significant reduction in arthritis as well as hind paw swelling scores. In terms of inflammation, HBP was considered to exert a more significant effect when combined with Enbrel^®^. Levels of serum inflammatory cytokines, including IFN*γ* and IL6, were also markedly increased in CIA mice and notably suppressed with Enbrel^®^. Unlike arthritis and paw swelling scores, serum cytokine concentrations showed limited response to co-injection of HBP and Enbrel^®^. The patterns of inflammatory cytokine concentrations in serum are reported to differ from arthritis and paw swelling scores in CIA mice, which may be attributed to the potential involvement of other factors in improving pro-inflammatory IFN*γ* or IL6 levels in blood. Further research is essential to clarify the specific pre- and pro-inflammatory mechanisms. Co-treatment of CIA mice with HBP and Enbrel^®^ exerted a marginally greater therapeutic effect on paw swelling relative to each single agent. While the results appear favorable, they may not be sufficiently beneficial to justify combination therapy in humans. Further research is required to establish whether Enbrel^®^ enhances the therapeutic efficacy of HBP in the clinic. 

We observed no apparent changes in organ weights (including liver, spleen, and kidney) in the HBP-treated group, compared with the normal control group (Appendix A). Moreover, no animal deaths were recorded during the experimental period, implying no cytotoxic and organ-specific toxic effects of HBP in vivo. 

In our experiments, HBP exerted similar anti-inflammatory effects to Enbrel^®^, even when used alone. The anti-inflammatory effect of the peptide appeared further enhanced upon co-treatment with Enbrel^®^. Additionally, injection of HBP led to recovery of the phenotype of inflammation-damaged chondrocytes. HBP-mediated chondrocyte recovery was confirmed based on reduced IL6 expression in cartilage tissue of treated mouse groups, which was accelerated with co-injection of HBP and Enbrel^®^ (Figure 8). Decreased expression of chondrocyte markers, including AGG, COLII, and TNFα, in LPS-stimulated human chondrocytes was reversed upon HBP treatment (Figure 2). Chondrocytes damaged via inflammation in the CIA animal model were also regenerated in the presence of HBP (Figure 6C). 

Our results collectively suggest that abnormal intracellular changes of joint chondrocytes in the CIA mouse model are effectively attenuated by HBP and/or Enbrel. In this regard, it may be advantageous to combine the peptide with Enbrel^®^ to optimize treatment of inflammation and recovery of joint chondrocyte structures in chronic arthritis. 

## 4. Materials and Methods 

### 4.1. Peptide Preparation

Heparin-binding peptides were generated using a peptide synthesizer (Prelude, Protein Technologies Inc., AZ, USA). The C-terminal amide form was produced using standard 9-fluorenylmethoxycarbonyl (Fmoc) chemistry. Rink amide-MBHA resin (GL Biochem, Shanghai, China) was pre-swollen in DMF (50 mg/mL) and Fmoc-protecting groups of resin and amino acids removed using 30% piperidine in DMF, 10 eq. DIPEA, 5 eq. HBTU, and 5 eq. Fmoc-protected amino acid, calculated according to resin loading. Cleavage and side-chain deprotection of the peptide resin was conducted for 4 h using a cleavage cocktail (TFA/water/thioanisole/phenol/ethanedithiol (8.5/0.5/0.5/0.5,0.25)). Solutions containing cleaved peptide were precipitated by the addition of chilled ether. Peptides were purified using preparative reverse-phase high-performance liquid chromatography (RP-HPLC; Waters, Milford, MA, USA) with a Vydac C18 column and 50 min gradient from 90% to 10% water/acetonitrile containing 0.1% trifluoroacetic acid (TFA). Peptide purity was determined as >98% via HPLC (Shimadzu, Kyoto, Japan) and liquid chromatography-mass spectrometry (LC-MS, Shimadzu, Kyoto, Japan). To establish the peptide translocation pathways, rhodamine was manually conjugated to the N-terminus during synthesis [49,50]. The purity and efficiency of rhodamine labeling was assayed via HPLC by monitoring absorbance at 230 nm and fluorescence at 440 nm. Fluorescence-labeled peptides were purified using HPLC (purity >95%), lyophilized, and stored at −20 °C in the dark until use.

### 4.2. Cell Culture

Human articular chondrocytes (NHAC, adult human knee chondrocytes) were purchased from Lonza (Walkersville, MD, USA) and used until passage 6. Cells were seeded onto 40 mm cell culture dishes (TPP, Trasadingen, Switzerland) at a concentration of 1 × 10^4^ cells/dish. Chondrocyte basal medium (Lonza) was supplemented with SingleQuots chondrocyte growth medium BulletKit^®^ (Lonza) composed of FBS, fibroblast growth factor 2 (FGF-2), gentamicin sulfate/amphotericin-B, insulin, insulin-like growth factor 1 (IGF-1), and transferrin. NHAC cells were sub-cultured two–three times to < 85% confluency and the medium changed twice a week.

Murine RAW264.7 macrophages were obtained from American Type Culture Collection (ATCC; Manassas, VA) and maintained in DMEM (Gibco-BRL, Grand Island, NY, USA) supplemented with 10% FBS (Gibco-BRL) and 1% antibiotic-antimycotic solution (Gibco-BRL) at 37 °C in a humidified atmosphere in 5% CO_2_.

### 4.3. MTT Assay

Cell-viability assay was analyzed by using an MTT assay. RAW264.7 cells were seeded in 96-well plates (1 × 10^4^ cells/well). After treatment with various concentrations of HBP (0, 0.01, 0.1, 1, 10, 100, and 200 μg/mL) for 24 h, the MTT solution was added to the cells and incubated for 4 h. After incubation, medium was removed and dimethyl sulfoxide (500 μL/well) was added to dissolve the formazan precipitates. Extracted formazan was transferred to a 96-well plate, and measured their absorbance at 570 nm using a microplate reader (BioTek Instruments, VT, USA). The cell viability assay was obtained from three independent experiments.

### 4.4. RNA Isolation and Quantitative RT-PCR

NHAC cells were plated onto 100 mm culture dishes and incubated at 37 °C under 5% CO_2_. At 80–90% confluence, cells were incubated for 2 h in DMEM with 0.5% serum for starvation, followed by treatment with LPS (1 μg/mL; St. Louis, MO, USA) for 24 h and various concentrations of HBP for 1 h.

The medium was replaced after 24 h with fresh medium containing HBP with or without LPS for a total culture period of 3 days. Total RNA from human articular chondrocytes was extracted using a TRIzol reagent according to the manufacturer’s instructions (Life Technologies, Darmstadt, Germany). Isolated RNA was treated with DNase I (Thermo Scientific, Schwerte, Germany) to remove possible genomic DNA contamination and used for cDNA synthesis with the aid of Superscript III Transcriptase (Life Technologies) and random hexamer primers (Thermo Scientific) at 50 °C for 1 h. Each sample was run in triplicate. Quantitative RT-PCR was conducted using POWER SYBR Green qPCR Master Mix (Life Technologies) and 0.2 μM primer (primer sequences are listed in Appendix A) on the StepOne Plus Real-Time PCR System (Applied Biosystems, Forster City, CA) under the following cycle conditions: primary denaturation at 95 °C for 5 min, 40 cycles of 30 s at 95 °C, 40 s at 60 °C, and 30 s at 72 °C, followed by fluorescence measurements.

### 4.5. In Vitro Cellular Internalization

RAW264.7 cells (1 × 10^4^ cells) were seeded on glass slides (Life Technologies, CA, USA). Following 24 h incubation to allow cell attachment, the culture medium was removed and 50 μM rhodamine-labeled HBP added along with fresh complete medium, followed by incubation for 10 min at 37 °C in a 5% CO_2_ atmosphere. After incubation, cells were washed with PBS and incubated for 30 min at room temperature with 1 μg/mL 4′,6-diamidino-2-phenylindole (DAPI) to visualize nuclei. Cells were washed with PBS and imaged using an Olympus FV-300 confocal laser scanning microscope operated with FLUOVIEW software (Olympus, Tokyo, Japan).

### 4.6. Western Blot Analysis

RAW264.7 cells were plated on 10 mm diameter culture dishes (1 × 10^5^ cells/dish). At 80–90% confluence, cells were incubated for 2 h in DMEM with 0.5% FBS for starvation [51]. Next, cells were treated with LPS (1 μg/mL) for 30 min or various concentrations of HBP for 1 h. Following cell lysis in RIPA containing protease and phosphatase inhibitors (Sigma, St. Louis, MO, USA) for 30 min on ice, total protein concentrations were determined with a Pierce BCA protein assay kit (Thermo Fisher Scientific, MA, USA). Equal amounts of protein (30 μg) were boiled in 5× electrophoresis sample buffer (0.25 M Tris-HCl, pH 6.8 15% SDS, 50% glycerol, 25% β-mercaptoethanol (ME), 0.01% bromophenol blue) for 5 min, separated via sodium dodecyl sulfate-polyacrylamide gel electrophoresis (SDS-PAGE) and electrotransferred onto nitrocellulose membranes. Non-specific protein binding was blocked via incubation with 3% BSA in T-TBS for 1 h. Membranes were washed with T-TBS and incubated with primary antibodies in T-TBS containing 3% BSA for 4 h at 4 °C, including anti-iNOS, COX2, IFN*γ*, and IL6 antibodies (Santa Cruz, CA, USA). After three further washes, membranes were incubated with a secondary antibody (horseradish peroxidase (HRP)-conjugated goat anti-rabbit IgG, diluted 1:2000 in 3% BSA) for 60 min. Blots were visualized with chemiluminescence reagents (Thermo Scientific, Schwerte, Germany). The relative optical densities of protein bands were quantified using Image J software (National Institutes of Health, Bethesda, MD, USA).

### 4.7. Induction of CIA in Mice and Peptide Treatment

Male DBA/1 (7 weeks old, 21–24 g) were bred at Harlan Co., Ltd. (Indianapolis, IN, USA) and supplied by Orientbio Inc. (Seungnam, Korea). All animal experiments were approved by the Institutional Animal Care and Use Committee of Chungbuk National University (IACUC ID: CBNUA-1095-17-02). Arthritis was induced via collagen inoculation (CIA) as described previously [43,52]. Mice were intradermally immunized with 500 μg collagen type II (COLII, Sigma) dissolved in 0.5 mL 0.1 M acetic acid and emulsified in 0.5 mL Freund’s incomplete adjuvant (Sigma) at 4 °C. Booster injections containing 250 μg COLII similarly dissolved and emulsified with Freund’s incomplete adjuvant (1:1) were administered intradermally into the tail base 21 days after primary immunization, following which hind paw volumes, survival, and body weights were monitored.

Mice were divided into five experimental groups as follows: normal control group (NT, *n* = 10) not immunized with collagen, CIA control group (RA-PBS, *n* = 10), HBP-treated CIA group (RA-HBP, 30 mg/kg, *n* = 10), Enbrel^®^ (Pfizer, Seoul, Korea)-treated CIA group (RA-E, 10 mg/kg, *n* = 10), and Enbrel^®^ and HBP co-injection group (RA-E+HBP, *n* = 10). At 28 days after primary immunization, HBP (30 mg/kg) was subcutaneously injected into the neck skin of CIA mice twice a week for 4 weeks. Enbrel^®^ was injected intraperitoneally three times a week for 4 weeks. The CIA control group was administered PBS. At the end of the experimental period, liver, spleen, and kidney weights were measured.

### 4.8. Assessment of Clinical Signs of Inflammation

Mice were monitored and absence of abnormal lesions confirmed for 28 days before the assessment of rheumatoid arthritis via visual evaluation of the limb joints. On the same schedule as HBP injection (days 0, 4, 7, 11, 14, 18, 21, and 25), clinical signs of inflammation were visually evaluated in the same manner and classified according to the clinical scoring system (arthritis score) as follows [53]: 0 (normal), 1 (slight swelling and/or erythema), 2 (pronounced edematous swelling), and 3 (Ankylosis). The average sum of the scores of the four limb joints was assessed through a blind test carried out by two trained investigators.

After visual evaluation, paw swelling was measured and thickness of the center of the sole of the limb measured using an electrical caliper (CD-15CPX, Kawasaki, Japan). The individual in charge of the measurement reduced the error by one person to the maximal extent and summed the measured values of edema at the same position in all four limbs.

### 4.9. Histological Examinations

On day 28, mice were killed for histological analysis. Both hind paws and ankles were harvested from each mouse and fixed overnight in 10% buffered formalin, decalcified in 30% citrate-buffered formic acid for 2 weeks at 4 °C, dehydrated in a graded series of methanol and xylene, and embedded in paraffin. Thin sections (5 μm) were stained with hematoxylin and eosin (HE) and histopathologic scoring performed under a light microscope by a blinded observer. The degree of inflammation around articular cartilage was analyzed according to a previously reported method and scored as follows: 0, mild; step 1, moderate; step 2, severe; step 3 [53].

### 4.10. Serum Cytokine Levels

At the end of the experimental period, serum samples were isolated from whole blood in each group following sacrifice via centrifugation at 4 °C for 30 min and stored at −80 °C until use. IFN*γ* and IL6 levels in serum were measured via ELISA (R&D Systems, Minneapolis, MN, USA) according to the manufacturer’s instructions.

### 4.11. Statistical Analysis

All data are presented as mean values ± standard deviation. Graphs were generated using GraphPad Prism 5 software (GraphPad Software Inc., La Jolla, CA, USA). Statistical significance was analyzed with one-way analysis of variance (ANOVA), followed by the Tukey’s post hoc multiple comparison test. Significant differences among experimental groups are indicated with different letters. In this study, we conducted three independent sets of in vitro experiments and used 10 mice per experimental group in vivo. All differences were considered significant at *p* values < 0.05.

## 5. Conclusions

Experiments from the current study demonstrated that a cell-penetrating HBP derived from the human heparin binding domain of BMP-4 could effectively suppress inflammation via regulation of iNOS-IFN*γ*-IL6 signaling in murine macrophages and human chondrocytes. Furthermore, the HBP reduced several arthritis symptoms, including hind paw swelling, chondrocyte inflammation, pro-inflammatory cytokine production, and increased arthritis score in CIA mice, and further protected against chondrocyte damage and bone structure at the joint by enhancing cartilage recovery through induction of IL6 expression. The data collectively support the utility of HBP as effective therapeutic agents or supplementary treatments for RA.

## Figures and Tables

**Figure 1 ijms-21-04251-f001:**
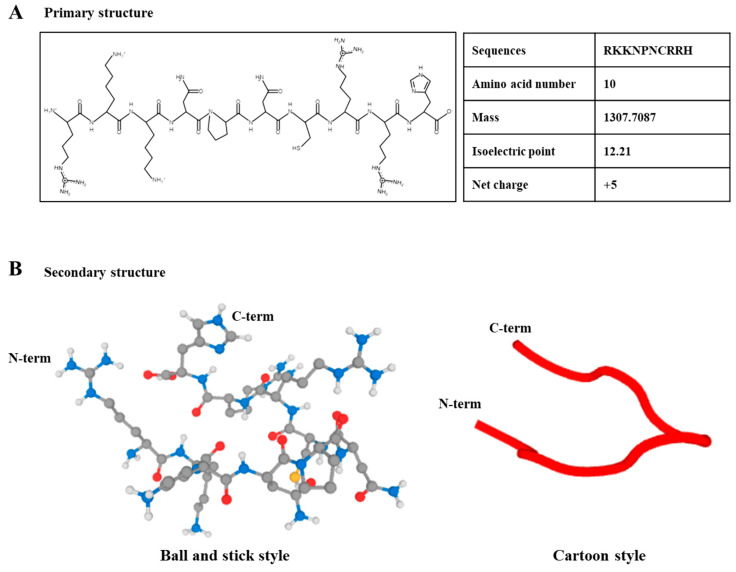
Structural illustration of heparin-binding peptide (HBP). (**A**) Primary structure of synthetic HBP composed of 10 amino acids and (**B**) secondary structure, presented as ball-and-stick and cartoon styles (Black; carbon, Blue; nitrogen, Red; Oxygen, Yellow; Sulfur; White; Hydrogen).

**Figure 2 ijms-21-04251-f002:**
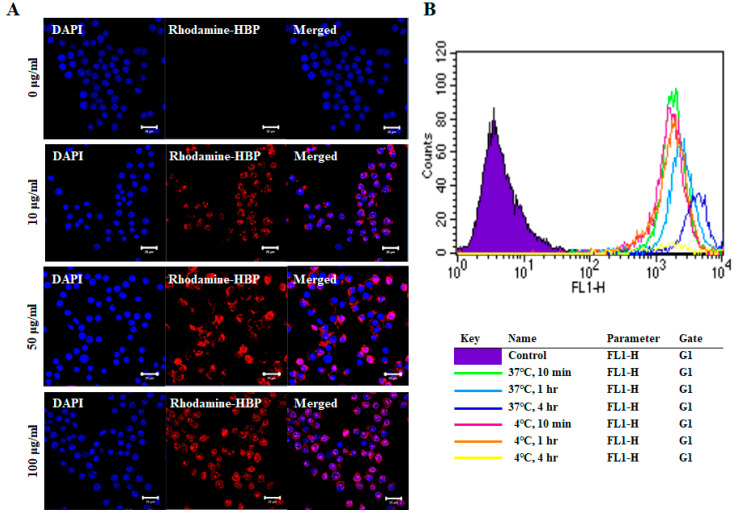
Images of intracellular HBP uptake in RAW264.7 cells. (**A**) Rhodamine-labeled HBP was detected in RAW264.7 cells via confocal microscopy. Intracellular HBP localization was increased in a dose-dependent manner (Scale bar = 20 µm, *n* = 3). (**B**) Fluorescence-activated cell sorting (FACS) analysis of differences in cellular uptake of HBP under different culture conditions, including temperature (4 °C, 37 °C) and incubation time (10 min, 1 h, and 4 h).

**Figure 3 ijms-21-04251-f003:**
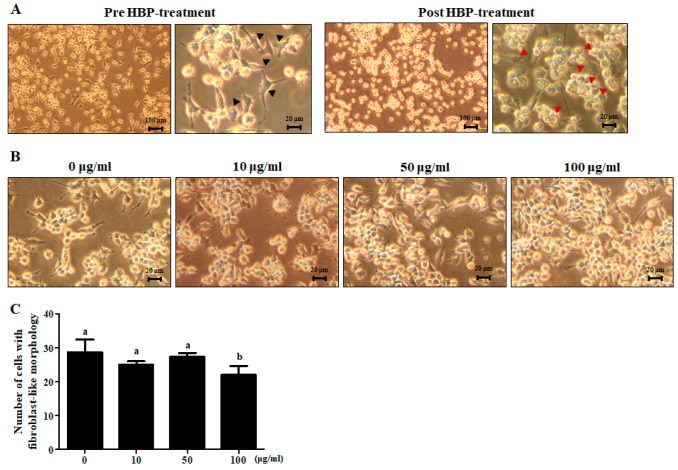
Light microscope view of morphological changes of lipopolysaccharide (LPS)-stimulated RAW264.7 and HBP treatment. (**A**) Cell morphology was examined before (left) and after HBP treatment (right) (*n* = 3). Black arrowheads signify LPS-stimulated inflammation of RAW264.7 cells. Red arrowheads represent RAW264.7 cells recovery following HBP treatment (magnification: ×40). (**B**) Morphology of LPS-stimulated RAW264.7 cells showing recovery following HBP treatment in a dose-dependent manner (magnification: ×200). (**C**) Bar graph indicating the number of cells showing fibroblast-like morphology.

**Figure 4 ijms-21-04251-f004:**
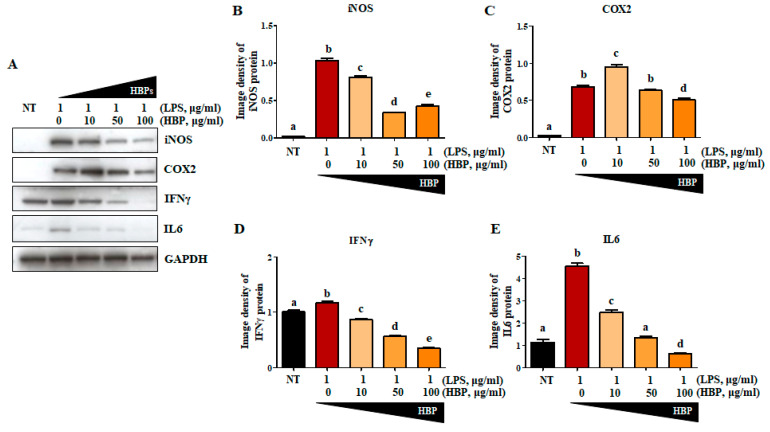
Effects of HBP on inflammatory marker protein expression in LPS-stimulated RAW264.7 macrophages. (**A**) Western blot analysis of inflammatory markers, including iNOS, COX2, IFN*γ*, and IL6, in LPS-stimulated RAW264.7 cells. (**B**)–(**E**) Band intensity of each protein (iNOS (**B**), COX2 (**C**), IFN*γ* (**D**), and IL6 (**E**)) presented as a bar graph normalized to the intensity of the corresponding GAPDH band (*n* = 3). Different alphabets (a, b, c, d, and e) in each Figure indicate significant differences among experimental groups (*p*  <  0.05).

**Figure 5 ijms-21-04251-f005:**
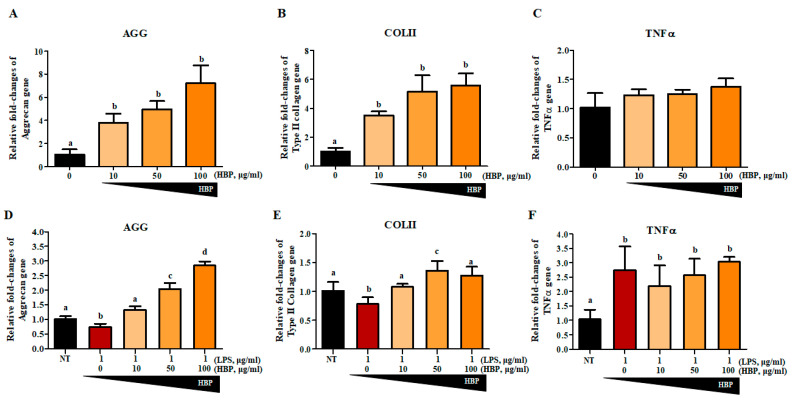
Gene expressions related to chondrocyte potentials with HBP treatment of human cartilage cells. The HBP itself increased (**A**) Aggrecan (AGG), (**B**) Collagen Type II (COLII), and (**C**) TNFα mRNA expressions in NHAC cells in a dose dependent manner (*p* < 0.05, *n* = 3). The LPS-stimulated were treated with various concentrations of HBP, followed by examination of cartilage regeneration-related gene expression. Expression changes in (**D**) AGG, (**E**) COLII and (**F**) TNFα were analyzed via quantitative PCR (*p* < 0.05, *n* = 3). Different alphabets (a, b, c, and d) in each Figure indicate significant differences among experimental groups (*p*  <  0.05).

**Figure 6 ijms-21-04251-f006:**
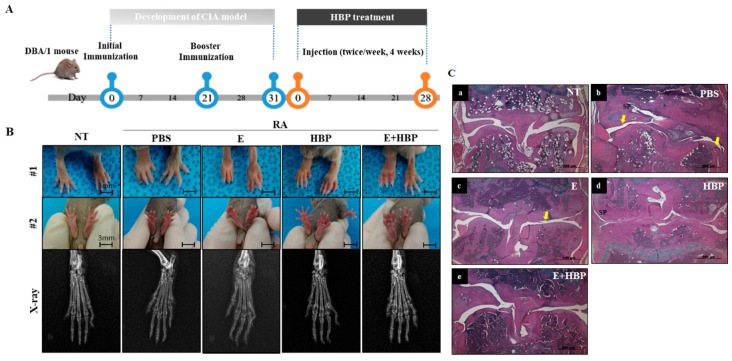
Development of a collagen-induced arthritis (CIA) mouse model and its application in examining the anti-rheumatic activity of HBP. (**A**) Schematic illustration of the induction of rheumatoid arthritis (RA) with collagen injection and HBP treatment. (**B**) Foot and x-ray images of all experimental groups are presented. (**C**) H&E staining of all experimental groups (SP; synovial proliferation, Yellow arrow; Pannus formation). (**D**,**E**) Changes in paw swelling scores over the entire experimental period. Results on day 28 from all experimental groups are presented in the bar graph (*n* = 10 per group). (**F**,**G**) Changes in arthritis scores throughout the entire experimental period (**F**). Results on day 28 from all experimental groups are presented in the bar graph (**G**). Statistical significance between groups was analyzed using the Student’s *t*-test. The level of significance is represented as * *p* < 0.05, ** *p* < 0.01, or *** *p* < 0.001.

**Figure 7 ijms-21-04251-f007:**
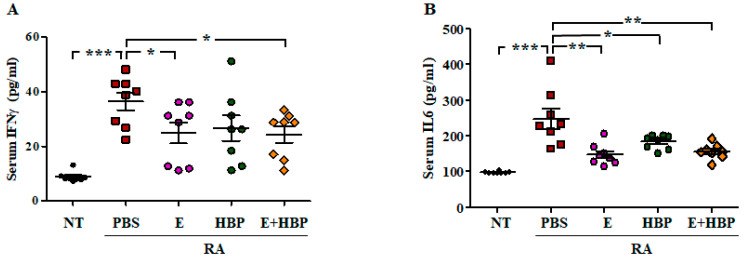
Comparison of serum cytokine levels among all experimental groups. Changes in serum IFN*γ* (**A**) and IL6 (**B**) concentrations in Enbrel^®^ and HBP-treated RA mice (*n* = 10 per group). Statistical significance between groups was analyzed using the Student’s *t*-test. The level of significance is represented as * *p* < 0.05, ** *p* < 0.01, or *** *p* < 0.001.

**Figure 8 ijms-21-04251-f008:**
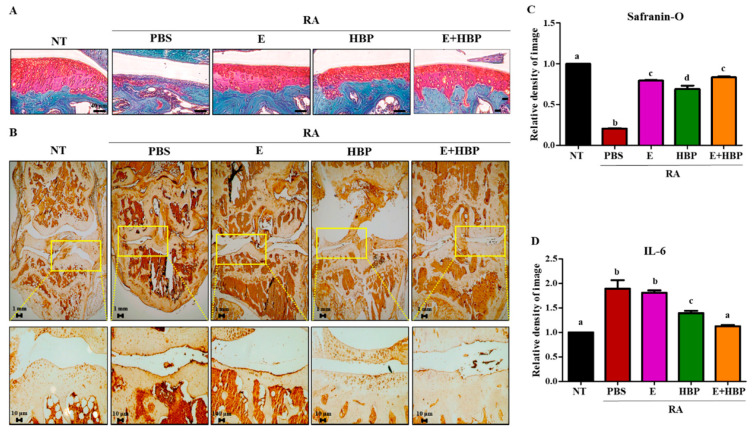
Effects of HBP on cartilage recovery determined via immunohistochemical analysis of IL6 expression and chondrogenic potential via Safranin-O staining. (**A**) Safranin-O-stained cartilage of RA mice. (**B**) IL6 protein in cartilage of RA mice was stained with avidin-biotin. (**C**) Quantification of the degree of Safranin-O and (**D**) IL6 staining of cartilage using Image J software, presented as bar graphs (*n* = 10 per group). Different alphabets (a, b, c, and d) in each Figure indicate significant differences among experimental groups (*p*  <  0.05).

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
