# Peer review of "A Synthetic Cell-Penetrating Heparin-Binding Peptide Derived from BMP4 with Anti-Inflammatory and Chondrogenic Functions for the Treatment of Arthritis"

_ijms, 2020, doi:10.3390/ijms21124251_

Round 1
Reviewer 1 Report
The resubmitted manuscript correctly addresses many of my comments in the first review.
The authors should be asked to acknowledge that the combination treatment with Enbrel and the HBP was not significantly better than either of the two agents used separately, as shown in the figures. P was not less than 0.05
The authors continue to use the terms regeneration and chondrogenic improperly. The effect of the HBP is simply anti-inflammatory.
Author Response
Point 1: The authors should be asked to acknowledge that the combination treatment with Enbrel and the HBP was not significantly better than either of the two agents used separately, as shown in the figures. P was not less than 0.05
Response 1:
We appreciate your comment and also agree that the significance of the combination effects was not significant. So, we changed the sentence as below;
(Line 233) CIA mice were co-treated with HBPs and Enbrel® to investigate their combined effects. The combination of peptide and Enbrel® exerted a marginally greater therapeutic effect on paw swelling than each single agent alone (Figure 6D, E).
(Line 373) CIA mice co-treated with HBPs and Enbrel® showed marginally greater therapeutic effect on paw swelling than each single agent alone. Although it represents the profitable results, it may not be sufficient to justify a combination therapy in humans. To empower the effect of HBP, it may need more time to present combinatory effect in clinic.
Point 2: The authors continue to use the terms regeneration and chondrogenic improperly. The effect of the HBP is simply anti-inflammatory.
Response 2:
We corrected the term regeneration to recovery. Since we found that the HBPs itself presented damaged cartilage tissue of RA mice by Safranin-O staining, we believe that the HBPs posses at least cartilage recovery effects.
(Line 190):
Figure 5. Gene expressions related to chondrocyte potentials with HBP treatment of human cartilage cells. The HBP itself increased (A) Aggrecan (AGG) and (B) Collagen Type II (COLII) (C) TNFα mRNA expressions in NHAC cells in a dose dependent manner (p < 0.05, n = 3). The LPS-stimulated were treated with various concentrations of HBP, followed by examination of cartilage recovery-related gene expression. Expression changes in (D) AGG, (E) COLII and (F) TNFα were analyzed via quantitative PCR (p < 0.05, n = 3).
(Line 260):
Although the HBP-only treated group showed significantly recovered chondrocyte recovery potential compared to PBS group, still it needs to be co-treated with an Enbrel®.
Reviewer 2 Report
The authors addressed my comments/concerns satisfactorily. However, the English language in the new (highlighted) parts should be carefully revised.
Author Response
Response 1:
The grammatical errors in the revised manuscript including yellow highlighted have been corrected severely by the help from English Manger (certificate is attached at page 20 of in manuscirpt).
